# Reduction in the Use of Some Herbicides Favors Nitrogen Fixation Efficiency in *Phaseolus vulgaris* and *Medicago sativa*

**DOI:** 10.3390/plants12081608

**Published:** 2023-04-10

**Authors:** Mario Paniagua-López, César Jiménez-Pelayo, Germán Orlando Gómez-Fernández, José Antonio Herrera-Cervera, Miguel López-Gómez

**Affiliations:** Department of Plant Physiology, Faculty of Sciences, Campus of Fuentenueva, University of Granada, 18071 Granada, Spain

**Keywords:** pendimethalin, clethodim, legumes, nitrogen fixation, symbiosis

## Abstract

In recent decades, the quality of agricultural soils has been seriously affected by the excessive application of pesticides, with herbicides being one of the most abundant. Continuous use of herbicides alters the soil microbial community and beneficial interactions between plants and bacteria such as legume-rhizobia spp. symbiosis, causing a decrease in the biological nitrogen fixation, which is essential for soil fertility. Therefore, the aim of this work was to study the effect of two commonly used herbicides (pendimethalin and clethodim) on the legume-rhizobia spp. symbiosis to improve the effectiveness of this process. *Phaseolus vulgaris* plants grown in pots with a mixture of soil:perlite (3:1 *v*/*v*), showed a 44% inhibition of nitrogen fixation rate with pendimethalin. However, clethodim, specifically used against monocots, did not induce significant differences. Additionally, we analyzed the effect of herbicides on root exudate composition, detecting alterations that might be interfering with the symbiosis establishment. In order to assess the effect of the herbicides at the early nodulation steps, nodulation kinetics in *Medicago sativa* plants inoculated with *Sinorhizobium meliloti* were performed. Clethodim caused a 30% reduction in nodulation while pendimethalin totally inhibited nodulation, producing a reduction in bacterial growth and motility as well. In conclusion, pendimethalin and clethodim application reduced the capacity of *Phaseolus vulgaris* and *Medicago sativa* to fix nitrogen by inhibiting root growth and modifying root exudate composition as well as bacterial fitness. Thus, a reduction in the use of these herbicides in these crops should be addressed to favor a state of natural fertilization of the soil through greater efficiency of leguminous crops.

## 1. Introduction

Conventional agricultural practices use agrochemicals, such as insecticides, fungicides, and herbicides to control crop pests and increase productivity [1]. Among the agrochemicals, herbicides, which are used to control weeds, constitute 50% of the total pesticides [2]. Pendimethalin (CAS nº 40487-42-1) is the third most used herbicide after glyphosate and paraquat belonging to the dinitroaniline class [3]. This herbicide is used for pre-emergence application to control grasses and dicotyledenous weeds in crops, such as cotton (*Gossypium hirsutum* L.), wheat (*Triticum aestivum* L.), soybean (*Glycine max* L.), maize (*Zea mays* L.), peas (*Pisum sativum* L.), and several vegetable crops. Pendimethalin has a relatively low volatility unlike other dinitroaniline herbicides and, therefore, dissipates less rapidly from the soil surface by volatilization [4]. In addition, this dinitroaniline herbicide is also less photodegradable than other dinitroanilines [5], which can cause its accumulation in agricultural soils [6,7] and aquatic ecosystems [8], affecting other nontarget organisms [9,10], including humans. In fact, pendimethalin has been classified as a possible human carcinogen [11,12] with negative effects on vasculature formation [13].

Clethodim (CAS nº 99129-21-2) is used for common bean (*Phaseolus vulgaris*) production [14] as a post-emergence herbicide for the control of monocotyledonous weeds, and its use has increased due to the development of resistance of some grasses to the most widespread herbicide, glyphosate. Clethodim is a cyclohexanedione oxime herbicide, highly water-soluble and poorly adsorbed in soil, which may move into water bodies as a potential contaminant [15,16] causing developmental toxicity in fishes [17]. Additionally, despite the photosensitivity of this compound, photodegraded solutions of clethodim have been shown to be more toxic than the herbicide itself, reaching the maximum toxicity when the herbicide is completely degraded [18].

The continuous application of herbicides might provoke accumulation in agricultural soils causing adverse effects on the soil microbiome [19,20,21]; this is of great importance considering the soil microbiome capacity to increase the accessibility of plants to nutrients, among other ecological functions [22,23]. Some of the soil bacteria, known as rhizobacteria, have the capacity to engage in stable associations with plant roots and establish symbiotic interactions.

One example of this is the association between legumes and soil bacteria known as rhizobia, with a capacity to fix atmospheric nitrogen that is supplied to plants [24]. The establishment of this interaction depends on the mutual recognition of the symbionts through chemical signals that are secreted by the plant roots into the rhizosphere [25,26]. Among these signals are flavonoids that induce responses in specific rhizobial bacteria, including changes in their growth and motility (chemotaxis), as well as the secretion of lipo-chitoligosaccarides, known as nodulation factors (NFs), specifically recognized by the root cells [27,28,29]. The specific recognitions between the plant cells and the bacteria lead to the formation of new organs in the plant roots called nodules where the bacteria, differentiated in bacteroids, are accommodated and acquire the capacity to reduce atmospheric nitrogen (N_2_) to ammonia. This reduced nitrogen is supplied to the plant, allowing it to become independent of an external nitrogen source [30,31]. This process known as biological nitrogen fixation has been a well-established practice for maintaining soil fertility in rotation with non-N-fixing crops until the introduction of chemical fertilizers during the second half of the 20th century [31,32].

Some reports suggest that herbicides have variable effects on the growth of rhizobia and, therefore, on the biological nitrogen fixation in agricultural systems [33,34,35,36,37]. Therefore, the potential toxic effect of pendimethalin and clethodim on rhizobia-legume symbiosis needs to be assessed in order to maximize the effectiveness of this plant-microbe association to increasesoil fertility through more sustainable crop management. Consequently, the effect of pendimethalin and clethodim has been analyzed in the symbiosis between *P. vulgaris* and *M. sativa* with their respective rhizobial strains in a pot trial and “in intro” assay to determine parameters related to the symbiosis establishment and efficiency, such as root exudate composition, nodulation, nitrogen fixation rate, and bacterial survival.

## 2. Results

### 2.1. Effectiveness of Herbicides

The analysis of the weed populations in the different treatments of the pot trial was carried out in order to understand the natural floristic composition of the soil used and the effectiveness of the herbicide treatments. The presence of six plant genera was observed in the control treatment: *Urtica*, *Lolium*, *Coniza*, *Amarantus*, *Convolvulus*, and *Stellaria*, all of which were considered weeds because they were not intentionally sown and were potential interferers in the growth and development of the species of interest. Weeds were discriminated according to the taxonomic criterion of *Liliopsida* (Monocotyledonae, *Lolium* only) and *Magnoliopsida* (Dicotyledonae) equivalent class, used as a more general method of differentiating herbicide target organisms. Statistical differences were determined based on the number of total individuals per class. Pendimethalin was effective in controlling all the plant genera observed since no individuals were observed in the pots treated with it (Figure 1). On the contrary, clethodim was only partially effective against Dicotyledonae plants, with no significant reductions, while it reduced more than 50% of the monocotyledonae plants of the genera *Lolium*.

### 2.2. Effect of Herbicides on Plant Growth and Nitrogen Fixation

Possible side effects of the herbicides on the biomass of *P. vulgaris* plants were analyzed as shown in Figure 2. No effects were detected in the shoot dry weight (SDW) for both herbicides, while a 25% reduction in the root dry weight (RDW) was observed in plants treated with pendimethalin.

Regarding nitrogen fixation parameters (Figure 3), no significant reduction was detected in the nodule number (NN); however, the pendimethalin treatment significantly reduced nodule fresh weight (NFW) (40%) and nitrogen fixation rate (NFR) (3.6 times), while no significant effect was induced by clethodim.

### 2.3. Nodulation Genes Expression

The expression of genes involved in early nodulation in roots of plants of *P. vulgaris* treated with clethodim and pendimethalin and inoculated with *Rhizobium tropici* was analyzed by RT-qPCR (Figure 4). Our analysis showed a reduction in the expression of *P. vulgaris ENOD40* nodulin gene, which is involved in nodule development, as well as in the expression of the subunit *PvNF-YA1* of the heterotrimeric nuclear factor Y (NF-Y), important for promoting both rhizobial infection and nodule development in *P. vulgaris* [38]. Only pendimethalin caused a significant reduction in the transcription of both genes, around half of the expression compared with the control plants. 

### 2.4. Root Exudates Composition

The effect of clethodim and pendimethalin on the composition of root exudates in terms of flavonoids and organic acids was analyzed in order to determine possible alterations in the recognition of the symbionts and, therefore, in the symbiosis establishment. The concentrations of different compounds were expressed per gram of root biomass, considering possible reductions of this parameter produced by the herbicides. Among the different flavonoids detected, naringenine and daidzein were the most abundant, followed by genisteine, apigenine, and hesperitine. In general, clethodim strongly reduced the presence of flavonoids in the root exudates: from 50% for hesperitin to 90% for genistein (Figure 5). Pendimethalin also inhibited the exudation of most of the flavonoids analyzed, although in that case, the inhibition range oscillated between about 70% for daidzein and around 30% for apigenin and naringenin. Hesperitin was the only flavonoid whose concentration increased 14-fold in response to pendimethalin, compared with control plants.

The content of organic acids in root exudates showed different responses for each herbicide (Figure 6), with no modifications produced by clethodim, while pendimethalin induced a strong increase in all of them that ranged from 18 times as much for citric acid to 5 times as much for tartaric acid. Indeed, succinic acid could only be detected in the root exudates of plants treated with pendimethalin. 

### 2.5. Nodulation Kinetics and Bacterial Growth and Motility

To determine the possible effects of the herbicides on nodule formation, nodulation kinetics were conducted in an in vitro nondestructive system that allows monitoring of nodule formation for several weeks. A not-significant reduction by clethodim in the nodule number could be observed for three weeks (Figure 7A). On the other hand, pendimethalin completely arrested nodulation.

To test the possible effects of the herbicides on bacterial growth and motility, growth curves were conducted under the herbicide concentrations used in the nodulation kinetics. Both herbicides inhibited bacterial growth with a significant reduction in the colony-forming units (CFUs) after 12 h. After 24 h (Figure 7B), only pendimethalin provoked a significant reduction in the CFUs. In addition to growth capacity, mobility is another trait required for bacterial root colonization. Therefore, bacterial mobility was tested (Figure 8), and swimming capacity was significantly reduced (by 35%) with the pendimethalin treatment, while clethodim did not produce significant changes.

## 3. Discussion

Agricultural intensification in most recent decades has strongly relied on chemical pesticides and fertilizers to improve crop productivity. However, despite a 10-fold increase in fertilizer production since the last century [39], the efficiency of N fertilization has declined due to the reduction in efficiency at higher levels of fertilizer application [40]. In addition to fertilizers, herbicides comprise 50% of total pesticides used globally [2] and continue expanding. Altogether, this results in a loss of soil organic matter and fertility [33], which drives increased application of chemical fertilizers and pesticides in an unsustainable manner. Therefore, sustainable agricultural practices to increase nutrient use efficiency and reduce environmental costs need to be implemented. Among such practices, the maximization of biological N fixation through the incorporation of legume crops and the enrichment of N-fixing rhizobia can increase the supply of organic and inorganic nutrients to the soil [41]. In this study, we have evaluated the effect of two commonly used herbicides on rhizobia-legume symbiosis with the aim to improve the effectiveness of this process in future research programs.

Pendimethalin provoked a reduction in the root biomass of *P. vulgaris* plants (Figure 2), which indicates a negative effect of this herbicide on the root initiation and development, as previously described in *Rhododendron obtusum* [42] and pigeon pea (*Cajanus cajan*) [43]. This negative effect on the root biomass could be due to the strong adsorption of this herbicide into the soil, driven by its high potential for hydrogen bonding and for its low-volatile and low-mobile behavior [4], as well as for the high organic matter content of the soil used, which is associated with increased soil binding capacity [6,7]. Root biomass reduction could affect the interaction with potentially beneficial microorganisms in the soil, including rhizobia. However, clethodim, which is targeted at grasses, did not affect plant biomass. In contrast to pendimethalin, clethodim is considered to rapidly dissipate in soil [33,44], mainly by photodegradation [18]; therefore, this herbicide induced fewer negative effects on plants compared with pendimethalin. The inhibition of the root growth produced by pendimethalin could explain the inhibition of symbiotic parameters, such as nitrogen fixation rate and nodule biomass (Figure 3), which were also detected in cowpea (*Vigna unguiculata*) and groundnut (*Arachis hypogaea*) plants [45,46]. We observed a significant reduction in nitrogenase activity (NFR), which indicates a decline in the metabolic activity in the nodule, as detected by Fox et al. [33] with different pesticides. Additionally, we found inhibition in the expression by pendimethalin of the nodulin genes *ENOD40* and *NF-YA1* (Figure 4), involved in the nodule development [38], which supports the reduction in the nodule biomass by this herbicide (Figure 3). 

Inhibition of nodulation by pendimethalin was confirmed in an in vitro experiment, in which the nodule number in *M. sativa* plants was monitored for three weeks. Pendimethalin completely inhibited nodulation (Figure 7A); this phenomenon could be related to a disruption of the initial chemical signaling between the plant and the bacteria, in which flavonoids, released in the root exudates, and Nod genes expression in the rhizobia are involved. In fact, inhibition of Nod gene expression in *S. meliloti* by pendimethalin has been previously described [47]. The inhibition of nodulation detected in *M. sativa* plants might be also produced by a negative effect on the growth and survival of *S. meliloti* exposed to the herbicides, which was analyzed in a growth curve and a motility assay (Figure 7B and Figure 8). In both assays, significant inhibition of bacterial survival and motility by pendimethalin was detected, which suggests that a reduction in bacterial fitness might be responsible for nodulation inhibition. In that sense, similar effects of pendimethalin on other rhizobia species have been previously described [48,49].

Several studies have shown that pesticides can alter root exudation [50,51] as a defense mechanism to enhance plant tolerance to pesticides [47] by different strategies that include (i) changes in soil pH to solubilize nutrients into assimilable forms, (ii) chelation of toxic compounds, (iii) attraction of beneficial microbiota, or (iv) releasing toxic substances for pathogens [52]. Therefore, we analyzed primary and secondary metabolites, such as organic acid and flavonoid concentrations in the root exudates of *P. vulgaris*, to determine their possible implications in the symbiotic alterations produced by the herbicides. The analysis of flavonoids showed a strong effect of both herbicides (Figure 5). However, flavonoid concentrations seem not to be critical for the recognition and nodule initiation since clethodim provoked a strong inhibition of flavonoid concentrations while it did not produce significant inhibition of nodulation in *P. vulgaris* and *M. sativa* plants. On the contrary, the effect of pendimethalin on the root exudate composition seems to affect nodulation to a greater extent. One of the most important modifications induced by pendimethalin was an increase in the hesperitin concentration (Figure 5). There are no previous reports about the specific implication of hesperitin in one of the strategies mentioned above, which deserves further investigation to understand the role of this flavonoid in response to pendimethalin. Organic acid exudation was also strongly increased by pendimethalin (Figure 6), which could be considered a symptom of stress, bearing in mind that under environmental stress, a greater amount of organic acids is released by plant roots into the rhizosphere [47]. Organic acids have been also described as important factors that regulate the degradation of persistent organic pollutants by attracting plant rhizosphere microorganisms [53]. The alteration of root exudate development could also be related to an alteration of root growth and development by stress conditions, affecting temporary concentrations of organic solutes at the root tips [52]. 

In conclusion, pendimethalin and clethodim applications reduced the legumes’ capacity to fix nitrogen by inhibiting root growth and nodule biomass and by modifying root exudate composition as well as bacterial fitness. Thus, a reduction in the use of these herbicides in crops should be addressed to favor a state of natural fertilization of the soil through greater efficiency of leguminous crops.

## 4. Materials and Methods

### 4.1. Plant Material and Growth Conditions

*Phaseolus vulgaris* (var. Contender) seeds were surface-sterilized by immersion in 5% NaClO plus Tween 20 (Sigma-Aldrich, Saint Louis, MO, USA) for 5 min and germinated onto wet, sterilized filter paper at 28 °C in the darkness. After 3 days, 2 seedlings were transferred to individual pots of 1.8 l filled with soil:perlite (3:1, *v*/*v*). Plants were grown in a controlled environmental chamber with a 16/8 h light–dark cycle, 26/22 °C day-night temperature, relative humidity of 55/75%, and photosynthetic photon flux density (400–700 nm) of 450 L mol m^−2^ s^−1^.

### 4.2. Soil Collection and Properties

The soil used was collected from an organic production farm, in the vicinity of the city of Granada, Spain, located at coordinates N 37°9′42.42.9″ W 3°33′23.889″, in January 2020 during the winter season. The soil was taken from the first 15 cm of the surface of a previously fallow plot. The soil was air-dried at room temperature in the laboratory, sieved at 2 mm, and homogenized. This fraction was used to characterize the main soil properties (Table 1). Soil texture was determined by the Robinson pipette method [54]; pH in a soil:water ratio 1:2.5 with a 914 pH/Conductometer Metrohm; soil:water extract (1:5) was prepared to determine the electrical conductivity using a Eutech CON700 conductivity meter (Oakton Instruments, Vernon Hills, IL, USA); available phosphorous was analyzed according to Olsen and Sommers [55]; finely ground samples were used for the measurement of calcium carbonate content by volumetric gases [56] and for the quantification of the organic carbon according to Tyurin [57]; and total nitrogen and total carbon were analyzed by dry combustion using an elemental analyzer (TruSpec CN LECO^®^, St. Joseph, MI, USA).

### 4.3. Herbicides Treatments and Harvest

Two commercial herbicides included in the list of authorized active substances of the European Union (Regulation (CE) No 1107/2009 of the European Parliament) and recommended for legume cultivation by the manufacturer were used. Centurion^®^ Plus (Bayer AG^®^) with clethodim as an active ingredient (AI) and Ordago^®^ (ADAMA Ltd.^®^, Madrid, Spain) with pendimethalin as an AI were applied following the recommended label dosage (Table 2). One liter of solution of each herbicide was prepared considering a water application volume of 600 L/ha. The dose per pot was adjusted based on the criterion of the surface area equivalent to 256 cm^2^ per pot. A two-liter garden sprayer with a full cone nozzle attached was used, and each product was applied mimicking field application conditions. The amount of solution applied per pot was equivalent to 1.56 mL. The applications of the non-control treatments (clethodim or pendimethalin) were carried out according to the conditions recommended by the respective manufacturers, considering variables such as the phenological stage of the plants and soil moisture. Therefore, as a pre-emergence herbicide, pendimethalin was applied 1 day after sowing while clethodim, used as a post-emergence herbicide, was applied 8 days after sowing. Pots were watered twice a week to keep 65% of the soil water holding capacity (WHC).

When the plants were at the vegetative growth stage and the symbiosis was well-established three weeks after sowing, the number and genera of individual weeds were quantified before the plants were harvested, and the nodules were detached and frozen at −80 °C for further analyses. Samples of shoots, roots, and nodules were weighed fresh and then dried at 70 °C for 24 h to determine their dry weight (DW).

### 4.4. Nitrogen Fixation

Nitrogenase activity (E.C. 1.7.9.92) was measured as the representative H_2_-evolution in an open-flow system [58] using an electrochemical H_2_ sensor (Qubit System Inc., Kingston, ON, Canada). H_2_ production was recorded on nodulated roots. Apparent nitrogenase activity (ANA, rate of H_2_ production in air) was determined under N_2_/O_2_ (80:20%) with a total flow of 0.4 L min^−1^. After reaching steady-state conditions, total nitrogenase activity (TNA) was determined under Ar/O_2_ (79:21%). Nitrogen fixation rate (NFR) was calculated as (TNA-ANA)/3. Standards of high-purity H_2_ were used to calibrate the detector.

### 4.5. Collection and Analysis of Root Exudates of Phaseolus vulgaris

To collect the beans root exudates (RE), germinated seeds were aseptically transferred to a hydroponic system consisting of a glass tube containing 15 mL of nitrogen-free nutrient solution [59] with pendimethalin and clethodim to a final concentration equivalent to the one used in the pot experiment (0.025 and 0.0025 µg mL^−1^, respectively). The plants were incubated aseptically for 5 days in a growth chamber with the same conditions as described before, protecting the roots from light by covering the tubes with black paper. On the fifth day, plants were removed from the tubes and RE and collected following Dardanelli et al. 2012 methodology [59], with some modifications, as follows: The nutrient solution was collected and centrifuged at 10,000 rpm for 20 min to remove root debris and microorganisms, and the supernatant containing the RE was passed through a C18 solid phase extraction cartridge (Restek Corp., Bellefonte, PA, USA). Flavonoids were eluted with 5 mL of 50%, 80%, and 100% methanol. The eluents were concentrated by lyophilization and stored at −20 °C. Exudates concentrated by lyophilization were dissolved in water and analyzed by chromatography as follows: For the organic acids, the plant roots were washed in 5 mL of deionized water for 24 h, this water was collected after centrifugation at 10,000 rpm for 20 min to remove root debris and was freeze-dried. The organic acids were analyzed using an ACQUITY H CLASS ultra-liquid chromatography coupled to a mass spectrometer detector (UPLC-MS) Xevo-TQ-XS (Waters, Milford, MA, USA) equipped with an electrospray ionization (ESI) source and a column Acquity UPLC HSS T3 2.1 × 100 mm, 1.8 µm. The column heater was set at 25 °C and the mobile phase flow rate was maintained at 0.4 mL/min. Mobile phases were performed using (A) water and (B) HPLC-grade acetonitrile (HiPerSolv) (Avantor, Radnor, PA, USA). The nonlinear separation gradient was 0–7 min: 0% B, T4: 0% B, T4.1: 100% B, T5: 100% B, T5.1: 0% B. The ESI operated in negative mode.

For the flavonoid analysis, the same equipment was used with a mobile phase composed of water plus 0.5% acetic acid (A) and eluent HPLC-grade acetonitrile (HiPerSolv) (Avantor, Radnor, PA, USA) (B). All aqueous solutions were prepared with Milli-Q water and vacuum-stripped and filtered using 0.2 μm membrane filters. The nonlinear separation gradient was 0–12 min: 5% B, T5: 50% B, T5.1: 100% B, T10: 100% B, T10.1: 5% B.

The contents of the individual organic acids and flavonoids were calculated using calibration curves of the corresponding standards purchased from Merk (Madrid, Spain) at concentrations ranging from 10 to 1000 ppb for flavonoids and 100 to 5000 ppb for organic acids.

### 4.6. Nodulation Kinetics and Bacteria Motility Assays

Nodulation kinetics were conducted with *Medicago sativa* plants grown axenically in glass tubes containing 10 mL of nitrogen-free nutrient solution [59]. Previously, seeds of *Medicago sativa* (var. Aragon) were surface-sterilized by immersion in 5% NaClO for 3 min and germinated in 0.8% water-agar plates at 25 °C in darkness. After two days, seedlings were transferred onto filter paper in the glass test tubes. One week after sowing, each seedling was inoculated with a *Sinorhizobium meliloti* strain 1021 cell suspension with an optical density at 600 nm wavelength of 0.03 (OD_600_ 0.03). At the inoculation time, a solution of clethodim or pendimethalin to reach a final concentration equivalent to the one used in the pot experiment (0.025 and 0.0025 µg mL^−1^) was added. The nodule number was monitored during the following 21 days. Growth curves of *S. meliloti* were performed in individual cultures in a rotary shaker at 100 rpm and 28 °C in 10 mL of TY medium (5 g/L tryptone, 3 g/L yeast extract, 0.9 g/L CaCl_2_) with clethodim or pendimethalin at the same concentrations as used for the nodulation kinetics. Starting inoculum contained approximately 10^6^ cells from overnight TY cultures. The assay was performed in triplicate for each treatment, and the evaluation of bacterial growth was performed by counting colonies after serial dilutions and plating on Petri plates containing TY media. Bacterial growth was verified at 2 different times (12 and 24 h), each with 3 biological replicates.

For bacterial motility, swimming assays were conducted in Petri dishes with 20 mL of 0.3% water-agar containing clethodim or pendimethalin to final concentrations equivalent to the concentration used in the pot experiment (Table 1). The plates were inoculated with *S. meliloti* by puncturing in the center cells from 1 mL of culture grown in TY broth to the late exponential phase (OD_600_nm = 1). The cells were washed two times with minimal medium and 10-fold concentrated. Every plate was inoculated on top with 2 µL of this bacterial suspension. After inoculation, plates remained open, and the culture drops were allowed to dry in a laminar flow hood for 10 min. Finally, plates were incubated at 28 °C for 7 days when surface motility was scored by measuring the diameter.

### 4.7. Nodulation Genes Expression Analysis

Plants of *Phaseolus vulgaris* were grown for 5 days in a hydroponic system consisting of a glass tube, as described previously for the root exudate collection, with pendimethalin and clethodim in a final concentration equivalent to the one used in the pot experiment (0.025 and 0.0025 µg mL^−1^, respectively). The seedlings were inoculated with 100 µL of *Rhizobium tropici* CIAT899 grown in TY broth to the late exponential phase (OD_600_nm = 1). On the fifth day, plants were removed from the tubes, and the roots were collected and frozen in liquid nitrogen. Total RNA was extracted using an RNeasy plant mini kit (Macherey-Nagel, Düren, Germany) followed by treatment with DNaseI (Ambion, Austin, TX, USA) for genomic DNA removal.

First-strand cDNA was reverse transcribed from 0.2 μg of DNase-treated total RNA. All DNase-treated total RNA samples were denatured at 65 °C for 5 min followed by a quick chill on ice in 12 μL of reaction mixture containing 25 pmol oligo-dT adapter primers and 1 μL of 10 mM deoxy-nucleotide triphosphate. After the addition of 4 μL of 5× First-Strand buffer, 1 μL of RNase OUT ribonuclease inhibitor, and 2 μL of 0.1 M dithiothreitol (DTT), the reaction was preheated at 37 °C for 3 min before the addition of 1 μL (200 U) of iScript reverse transcriptase (Bio-Rad, Hercules, CA, USA). The reaction mixture was incubated at 42 °C for 50 min, followed by heat inactivation at 70 °C for 15 min. The resulting first-strand cDNA was amplified using gene-specific primers provided by [60] (Appendix A).

Real-time PCR (qPCR) was performed using an SYBR Green PCR Master kit containing the Platinum Taq DNA polymerase (Thermo Fisher, Waltham, MA, USA) on an iCycler IQ thermocycler (Bio-Rad). The PCR amplification mixtures contained 10 ng of template cDNA and 0.5 U of polymerase. The cycling conditions were chosen according to the manufacturer’s instructions. They comprised 10 min polymerase activation at 95 °C and 35 cycles at 95 °C for 15 s, 55 °C for 30 s, and 72 °C for 20 s. Results were quantified with the DDCt method [61]. Transcript levels were normalized to ubiquitin gene expression. The primer sequences for qPCR analysis are provided in Appendix A.

### 4.8. Statistical Analysis

Experiments were arranged in a completely randomized design with 12 plants per treatment in the pot trial and the root exudate assays, 30 plants in the nodulation kinetic, and 6 plants grouped in 3 replicates in the nodulation gene expression experiment. The data were subjected to a one-way analysis of variance (ANOVA) with herbicides treatment (untreated control, clethodim, or pendimethalin) as the variable factor using the IBM SPSS Statistics 28.0.0.0 software, and significant differences among treatments were determined by Tukey’s post hoc test (*p* ≤ 0.05). Levene’s test was used to check the homogeneity of variances. Mean values ± SE error bars are represented.

## Figures and Tables

**Figure 1 plants-12-01608-f001:**
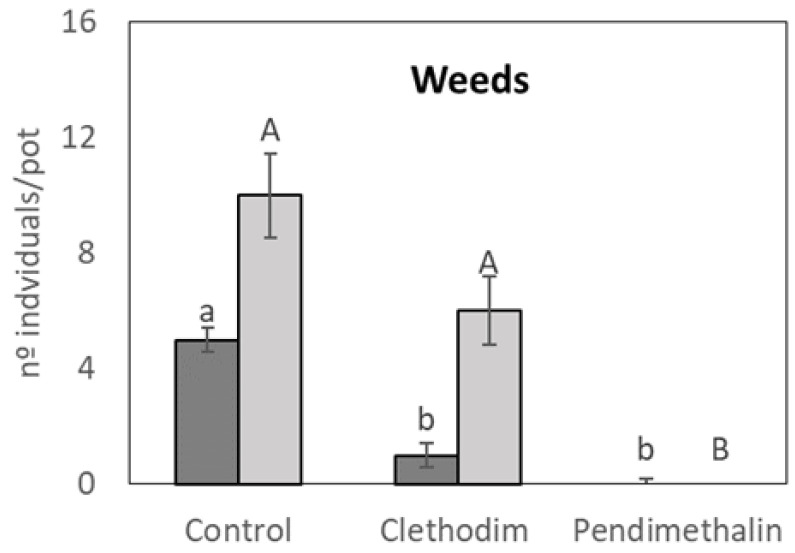
Individual number of weeds per pot, differentiating between monocotiledoneae (dark grey bars) and dicotiledoneae (clear grey bars) in a controlled chamber study in Granada, Spain, in January 2020. Data are mean + SE (*n* = 6). Mean values followed by similar lowercase (monocotyledoneae) or capital (dicotyledoneae) letters do not differ (*p* ≤ 0.05) using Tukey’s post hoc test.

**Figure 2 plants-12-01608-f002:**
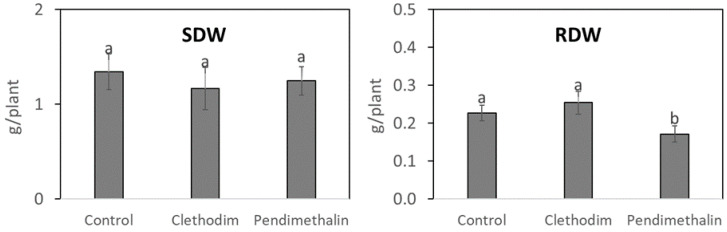
Shoot dry weight (SDW) and root dry weight (RDW) of *P. vulgaris* plants treated with clethodim and pendimethalin individually in a controlled chamber study in Granada, Spain, in January 2020. Data are mean + SE (*n* = 6). Mean values followed by the same letter do not differ (*p* ≤ 0.05) using Tukey’s post hoc test.

**Figure 3 plants-12-01608-f003:**
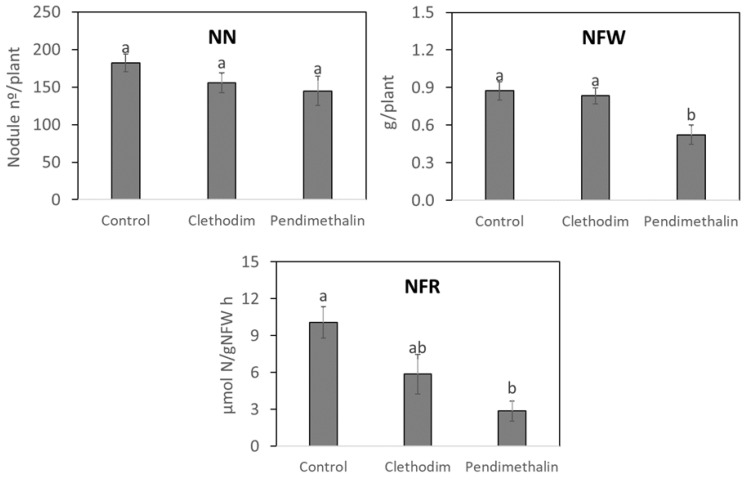
Nodule number (NN), nodule fresh weight (NFW), and nitrogen fixation rate (NFR) of *P. vulgaris* plants treated with clethodim and pendimethalin individually in a controlled chamber study in Granada, Spain, in January 2020. Data are mean ± SE (*n* = 6). Mean values followed by the same letter do not differ (*p* ≤ 0.05) using Tukey’s post hoc test.

**Figure 4 plants-12-01608-f004:**
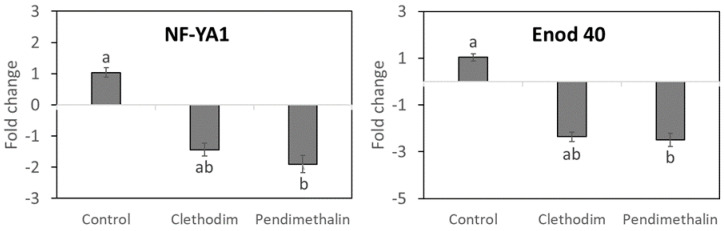
Expression of *P. vulgaris* genes related to rhizobial infection (NF-YA1) and early nodulation in roots (Enod 40) inoculated with *Rhizobium tropici*. *P. vulgaris* plants treated with clethodim and pendimethalin individually in a controlled chamber study in Granada, Spain, in January 2020. The ubiquitin UBC9 was used to normalize gene expression. When ratio values were lower than 1, the inverse value was estimated, and the sign was changed. Data are the mean ± SE (*n* = 3) in a single qRT-PCR experiment. Mean values followed by the same letter do not differ (*p* ≤ 0.05) using Tukey’s post hoc test.

**Figure 5 plants-12-01608-f005:**
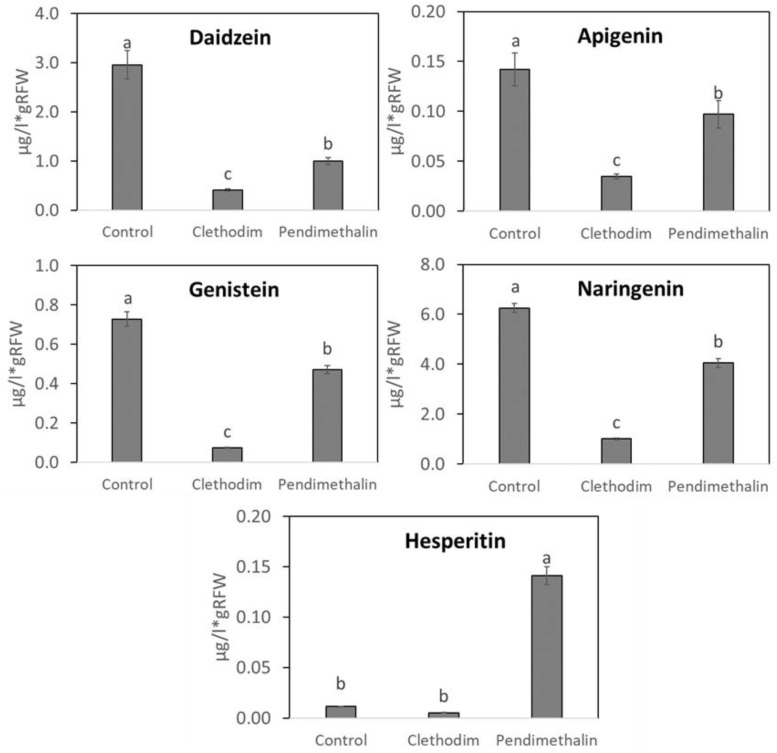
Content of flavonoids in root exudates of *P. vulgaris* plants treated with clethodim and pendimethalin individually in a controlled chamber study in Granada, Spain, in January 2020. RFW = root fresh weight. Data are mean ± SE (*n* = 6). Mean values followed by the same letter do not differ (*p* ≤ 0.05) using Tukey’s post hoc test.

**Figure 6 plants-12-01608-f006:**
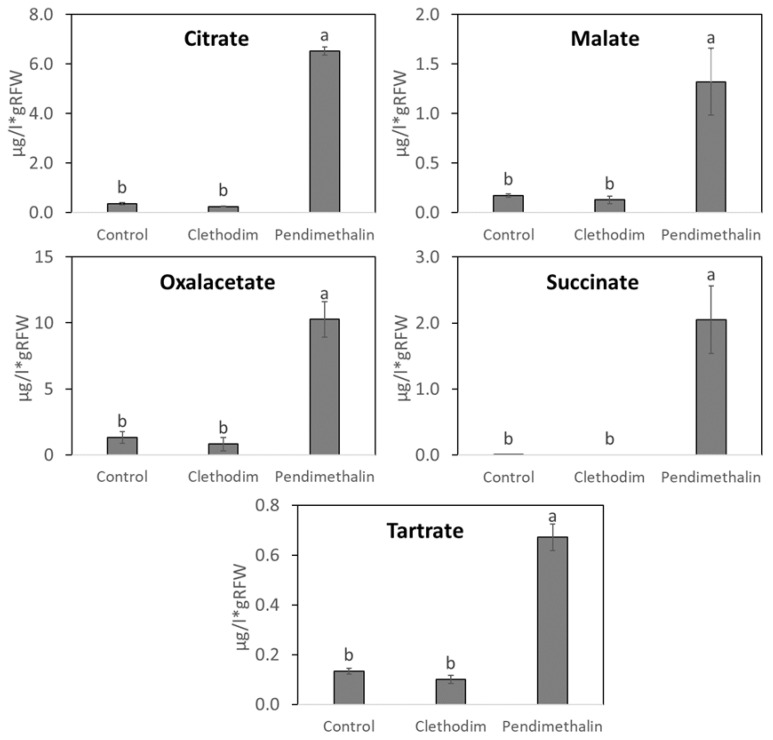
Content of organic acids in root exudates of *P. vulgaris* plants treated with clethodim and pendimethalin individually in a controlled chamber study in Granada, Spain, in January 2020. RFW = root fresh weight. Data are mean ± SE (*n* = 6). Mean values followed by the same letter do not differ (*p* ≤ 0.05) using Tukey’s post-hoctest.

**Figure 7 plants-12-01608-f007:**
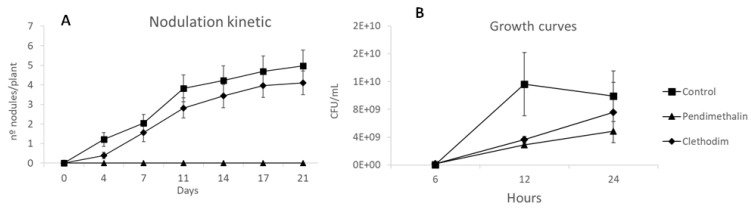
(**A**) Nodulation kinetic of *Medicago sativa* plants inoculated with *Sinorhizobium meliloti* (1021) treated with Clethodim and Pendimethalin in an in vitro study at Granada, Spain, in 2020. Data are mean + SE (*n* = 30). (**B**) Growth curves of *Sinorhizobium meliloti* (1021) grown in TY medium containing clethodim and pendimethalin. CFU = Colony-Forming Units. Data are mean ± SE (*n* = 3).

**Figure 8 plants-12-01608-f008:**
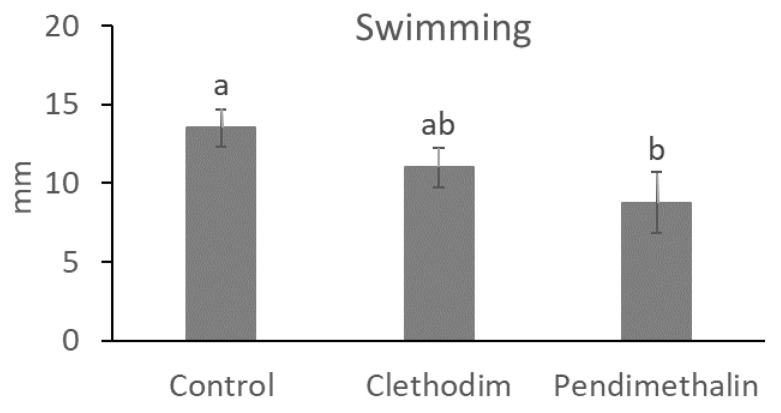
Swimming motility diameter of *Sinorhizobium meliloti* (1021) growth with clethodim and pendimethalin in 0.3% water-agar medium in an in vitro study at Granada, Spain, in 2020. Data are mean ± SE (*n* = 3). Mean values followed by the same letter do not differ (*p* ≤ 0.05) using Tukey’s post hoc test.

**Table 1 plants-12-01608-t001:** Soil properties: electrical conductivity (EC); total N (N) and total C (C); organic C (OC); calcium carbonate (CaCO_3_); available P (Pav); soil texture (Clay, Silt, Sand; Appendix A).

pH	EC (ds m^−1^)	N (g kg^−1^)	C (g kg^−1^)	OC (g kg^−1^)	CaCO_3_ (g kg^−1^)	P_av_ (mg kg^−1^)	Clay (%)	Silt (%)	Sand (%)
7.42	0.79	2.4	52.2	20.5	15.7	51.0	30.9	14.8	54.3

**Table 2 plants-12-01608-t002:** Detailed information on the herbicide products used.

Herbicide	Classification HRAC *	Product Name	Manufacturer	Concentration Active Ingredient (AI)	Product Dosage	Product Amount per Pot
Pendimethalin	Cycloexanodiona	Ordago^®^	ADAMA (Spain)	40% p/v	3 L/ha (1.2 kg AI/ha)	3.07 µg
Clethodim	Dinitroanilina	Centurión^®^	Bayer (Switzerland)	12% p/v	1 L/ha (0.12 kg AI/ha)	0.30 µg

* HRAC: International herbicide-resistant weed database (http://www.weedscience.org/Pages/aboutus.aspx) (accessed on 4 April 2023).

## Data Availability

The data presented in this study are available on request from the corresponding author.

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
