# Peer review of "Reduction in the Use of Some Herbicides Favors Nitrogen Fixation Efficiency in Phaseolus vulgaris and Medicago sativa"

_plants, 2023, doi:10.3390/plants12081608_

Round 1

Reviewer 1 Report (Previous Reviewer 1)

Line 74:change wording: “employ” to “use”

Lines 515-516: delete “reductions which ranged”. The sentence should read “reduced the presence of flavonoids in the root exudates by 50% for hesperitin to 90% for genistein”…

Line 523: increase, not increased

Line 550: RFW = Root Fresh Weight

Line 624: same correction as in line 550

Line 637: Colony Forming Units

Line 778: same correction

Line 794 in vitro, not in vitro

Author Response

Thanks for the comments,

in the present version of the manuscript all the corrections have been considered.

Reviewer 2 Report (Previous Reviewer 2)

The manuscript is much improved. Thanks. I only found a few items needing revision:

Lines 50-51: Delete the text following the corresponding author's email address, including the semicolon ';'

Line 85: Replace 'provoke' with 'can cause'

Line 86: 'organisms'

Line 168: 'weeds and usage has increased'

Line 449: Replace 'common bean' with 'P. vulgaris'

Line 801: 'of fertilizer production' (no 's' on fertilizer).

Lines 888-889: 'pigeon pea (Cajanus cajan)'

Line 901: 'groundnut'

Line 904: 'Additionally, we found' ('also' is redundant to 'additionally')

Line 925: 'organic acid and flavenoid concentrations'

Line 941-1173 (check continuity of line numbers across pages): 'of root exudate development'

Line 1365: 'individual weeds'

Line 1587: Replace 'freeze' with 'frozen'

Line1589: Replace 'Texas' with 'TX' remove the comma between 'TX' and 'USA'

Line 1598: Replace 'resulted' (no such word) with 'resulting' or 'resultant'

Line 1602: 'The PCR'

Thank you for adding references from mainline scientific literature!

Author Response

Thanks for the comments,

in the present version of the manuscript all the corrections have been considered.

This manuscript is a resubmission of an earlier submission. The following is a list of the peer review reports and author responses from that submission.

Round 1

Reviewer 1 Report

1.       The Aa correctly quote some references (e.g. ref 19) without discussing it later in the text. I suggest to put in more evidence the novelty of your study with respect to existing knowledge

2.       Line 352: did not result

3.       The final overall conclusion refers to field, agronomic conditions. However the study is carried out in vitro controlled conditions. At least the in vitro results on nodulation and ARA test for dinitrogen fixation should have been carried out in the field.

Reviewer 2 Report

Comments to Authors: MDPI Plants-2155723, Reduction in the use of herbicides favors nitrogen fixation efficiency in legumes

The manuscript is fairly well-written and organized and the English grammar only needs minor improvement. Thank you! The topic is of international interest; however, there are major concerns regarding how the research was conducted that, although possibly not fatal to the research, must be addressed before the manuscript will be ready for publication.

1.      The herbicides selected for the study represent a very small portion of the total population of available crop protection products, although the authors attempt to extrapolate their information across the entire population.

2.      While  the authors state that the label for the products used was followed in regard to application amount and timing, they apparently did not review labels for other countries (MDPI Plants is an international journal) to learn about requirements regarding timing and application for the selected products and crops were that would lead to crop protection. Specifically, pendamethalin is still not labeled for Medicago sativa in New Zealand (https://www.google.com/url?sa=t&rct=j&q=&esrc=s&source=web&cd=&cad=rja&uact=8&ved=2ahUKEwiWx9LYhb78AhUnmGoFHZ98D_4QFnoECA4QAQ&url=https%3A%2F%2Fagpro.co.nz%2Fdownloads%2Flabel-agpro-pendimethalin-ioizc.pdf&usg=AOvVaw3yRojYqCMkSuPp2k1kuhH-), while in the US, pendamethalin is forbidden to be applied before Medicago sativa reaches the second trifoliate stage (http://www.cdms.net/LabelsSDS/home#), which would not occur within 7 days after germination.

3.      It also is obvious that the authors have a bias against the use of pesticides. Any bias is contrary to scientific research for the advancement of knowledge. The study could have been conducted and reported without such a heavy emphasis on environmental concerns that do not pertain to crop safety and the symbiotic relationship. Lines 61-62 acknowledge that such problems occur when herbicides are applied indiscriminately, which appears to be what the authors did in not researching labels to see what governmental organizations in other countries require regarding the use of the herbicides selected for the study.

4.      Check literature in more internationally recognized journals regarding crop safety studies for the herbicides tested.

Otherwise, line-by-line comments are provided below for improvement in hopes that somehow the authors can salvage their efforts put into the research:

Title: The title should reflect that only two herbicides and two legumes were evaluated. I recommend something like: “Reduction in the use of some herbicides may favor nitrogen fixation efficiency in Medicago sativa and Phaseolus vulgaris

Line 5: Translate the contact information into English.

Line 10: ‘. . . pesticides, herbicides being one . . .’

Line 25: ‘. . . use of these herbicides in these crops . . .

Line 32 and throughout: Reference numbers are to be enclosed in [] rather than ().

Line 33 and elsewhere: ‘establish’ not ‘stablish’

Line 37 and many other places: Commas are being inappropriately. There really are a lot of instances of this.

Line 41: ‘The specific recognitions’

Line 45: ‘plant allowing it to become’

Line 46: ‘well-established’

Line 48: Begin new paragraph with the sentence that starts on this line.

Lines 50-51: ‘. . . crop pests and increases productivity [12]. Herbicides that are used to control weeds worldwide constitute 50% . . .’ Note, no comma after ‘worldwide’

Lines 52 & 64: Whenever a herbicide product is named, show the chemical name from the label in parentheses () or brackets [] immediately following.

Line 53: ‘. . . class [14]. It is used . . .’

Line 61: ‘Indeed, some reports suggest that . . .’

Line 66: ‘including common bean (Phaseolus vulgaris) [23].’

Line 68: ‘. . . contaminant [24].’

Line 75: ‘P. vulgaris’ Once an organism is identified by genus and species, only the first letter of the genus is needed until another genus is name that begins with the same letter.

Line 76: ‘in vitro

Results: How do your results for the control treatment, particularly SDW, RDW, nodule number and weight, compare to that measured by others for the same species at the same plant age. This is necessary to report to indicate that your data is comparable to what has been measured elsewhere without treatment effects.

Line 81: ‘. . . the weed populations in . . .’ This measurement was not described in the methods section. All reported data must be described in the methods.

Lines 83-84: ‘The presence of six plant genera was observed . . .’

Line 85: ‘Stellaria, all of which considered . . .’

Line 87: ‘. . .(Monocotyledonae, Lolium only) and . . .’ This would merely be for clarification that Lolium was the only genera of  Monocotyledonae found in the study.

Lines 108-110 (Figure 1 caption; changes underlined, etc.): ‘Individual number of weeds per pot differentiating between monocotyledonae (dark grey bars) and dicotyledonae (clear grey bars) in a controlled chamber study at Granada, Spain in January 2020. Data are mean + SE (n=6). Mean values followed by similar lower case (monocotyledonae) or capital (dicotyledonae) letters do not differ (p≤0.05) using the Duncan test.’ Always show the location of the study as well as the nature of the study (e.g., field, greenhouse, controlled chamber) in each table title or figure caption. Wording about letters was changed because ‘ab’ is used in some figures and to limit it to different letters would mean that a and ab were different and ab and b were different, but that’s not the intent. Apply these changes to subsequent figures and the table, where appropriate.

Line 114-115: shoot dry weight (SDW) . . . root dry weight (RDW

Lines 118-120: ‘. . . nodule number (NN) . . .’, ‘. . . nodule fresh weight (NFW) . . .’, ‘. . . N fixation rate (NFR), . . .’ Do define abbreviations on first use in the abstract and the main text, and tables and figures (Instructions to Authors/manuscript template says on first use in tables and figures, but it is good to define in all tables and figures so they can completely and individually stand alone from the text. Most of the time the methods come before the results and abbreviations defined in the methods cover the requirement of first mention, but MDPI Plants places the methods last. So, the abbreviations should be defined earlier for use in the remainder of the main text. Additionally, you state that no NN reduction was detected, but then tried to imply a reduction ‘(20% with pendimethalin)’ That is not appropriate. Also, clethodim reduction is nearly the same, but you did not emphasize that.

Line 133 (Figure 2 caption and y-axes): ‘Shoot dry weight (SDW) and root dry weight (RDW) of P. vulgaris plants treated with clethodim or pendimethalin in a controlled chamber study at Granada, Spain in 2020. Data are mean + SE (n=6). Mean values followed by similar letters do not differ (p≤0.05) using the 134 Duncan test.’ ‘and’ between herbicide names means that both were applied as the treatment when the indication is that they were applied individually as separate treatments. Check this also in the main text. Remember to add the location and type of the study. In the y-axes and throughout the manuscript, replace commas with decimals in decimal fractions.

Lines 150-152 (Figure 3 caption): ‘Nodule number (NN), nodule fresh weight (NFW) and nitrogen fixation rate (NFR) of P. vulgaris plants treated with clethodim or pendimethalin in a controlled chamber study at Granada, Spain in 2020. Data are mean + SE (n=6). Mean values followed by similar letters do not differ (p≤0.05) using the Duncan test.’ Note the unmarked deletion of some text.

Line 163: ‘from 50% for hesperitin to 90% for genistein . . .’ Decreases are generally not described as a ‘2x’ reduction.

Line 165: ‘. . . between about 70% for daidzein and around 30% . . .’

Line 166 and elsewhere: Replace ‘incremented’ with ‘increased’

Line 170: ‘. . . strong increases of all that ranged . . .’

Lines 209-211 (Figure 4 caption): ‘Content of flavonoids in root of P. vulgaris plants treated with clethodim or pendimethalin in a controlled chamber study at Granada, Spain in 2020. Data are mean + SE (n=6). 210 Mean values followed by similar letters do not differ (p≤0.05) using the Duncan test.’ Note the unmarked deletion of some text.

Lines 262-264 (Figure 5): ‘Content of organic acids in root exudates of P. vulgaris plants treated with clethodim or pendimethalin in a controlled chamber study at Granada, Spain in 2020. RFW signifies root fresh weight. Data are mean + SE (n=6). Mean values followed by similar letters do not differ (p≤0.05) using the Duncan test.’ I recommend removing the ‘ac.’ from each pane. The title states that they are acids. Remember to change commas to decimals in the y-axes values.

Lines 256-257: ‘. . reduction in the colony forming units (CFUs) after 12 h.

Line 271: ‘A slight, but non-significant reduction . . .’ Since it is non-significant, it is not a reduction. Be very careful about ‘stretching’ your results.

Lines 272-273: Was there a significant difference between the control and pendimethalin? If so, show Duncan results. You do not describe and regression analysis, or it would be appropriate to show the regression formulae and r-values. If there was no difference, admit that there was no difference in nodule numbers over time and delete the figure as would be misleading. If there was a significant difference, you must acknowledge that your application timing was not consistent with label requirements for pendimethalin in other countries where crop protection is more stringently controlled. Nowhere does any label say that you can immerse roots in the herbicide solution AND, as previously mentioned, you applied the herbicide before the alfalfa had a chance to nodulate for become well-enough established. In the US, that would be a violation of federal law that can carry fairly stiff penalties.

Lines 296-299 (Figure 6): ‘Figure 6. A. Nodulation kinetic of Medicago sativa plants inoculated with Sinorhizobium meliloti (1021) treated with clethodim or pendimethalin in an in vitro study at Granada, Spain in 2020. Data are mean + SE (n=30). B. Growth curves of Sinorhizobium meliloti (1021) grown in TY medium containing clethodim and pendimethalin. CFU signifies colony forming units. Data are mean + SE (n=3).’ Use of both solidus (/) AND a negative superscript to indicate division within the SI unit (CFU/ml-1). First, this indicates a multiplication of CFU * ml. Second, there is no consistency in the manuscript, including figures about which sign to use. Either use a solidus only or use the negative superscript only.Add the letters to the panes. Spell out the treatment names in the legend (Pendimethalin and Clethodim).

Lines 314-316: ‘Swimming motility diameter of Sinorhizobium meliloti (1021) growth with clethodim or pendimethalin in 0.3% water-agar medium in an in vitro study at Granada, Spain in 2020. Data are mean + SE (n=3). Mean values followed by similar letters do not differ (p≤0.05) using the 315 Duncan test.’

Lines 320-321: ’. . . 10-fold increase in fertilizer production . . ’

Line 321-322: ‘. . .fertilization has declined . . .’

Lines 324-326: This is an incomplete sentence. Perhaps revise to ‘Altogether, these result in . . .’

Line 329: Change ‘increment’ to ‘increase’

Line 333: ‘Pendimethalin provoked . . .’ Omit ‘A broad spectrum herbicide like’ It was pendimethalin and you did not test other such products. Stay within the scope of the study. Also, keep in mind that the product was not applied in a manner found to be safe in other countries.

Line 334: ‘indicates’

Line 336: Insert the scientific name for pigeon pea (no capital letters) following ‘pigeon pea’

Lines 336-340: The negative effect was likely due to incorrect application timing for the crop at the particular stage of maturity (the US label says to plant the crop 5 cm deep and surface apply the pendimethalin within 1-4 days after planting, not 1-4 days after germination; New Zealand does not label pendimethalin for beans, only peas).

Line 340: ‘soil binding capacity [32]. Root biomass reduction . . .’

Line 343: ‘affect’

Line 344: Replace ‘lower’ with ‘less’

Line 347: ‘. . . cowpea (scientific name) and groundnut (scientific name) plants . . .’

Line 350: ‘transforms’

Line 35-354: This sentence is long and incomplete. Subdivide and form complete statements from each sentence, perhaps like: ‘Nitrogenase activity declined with both herbicides, although only pendimethalin provoked a significant inhibition (data not shown). This was contrary to the findings of Fox et al. [38] who did not observe a significant reduction of the plant yield, which would have probably been observed at later pheno-353 logical stages affecting pods and grain seeds productivity.’ A citation is required to support the statement about ‘probable effects observed at later stages’ But, when the herbicide is correctly applied, plants suffering non-lethal chemical damage will generally recover and outgrow the effects to minimize yield loss. Otherwise, the product would have little or no market demand. Also, while the methods states that nitrogenase activity was measured, it was not reported in the results section. Base your discussion on the results that are reported and call out the figure showing the results or state, ‘data not shown’ parenthetically.

Line 355: According to lines 271-273, the effect was non-significant for clethodim.

Lines 357-360: It probably did disrupt that initial signaling, but this does not occur in the US when the label is followed because of planting and application timing guidelines that are part of the label. In the US pesticide labels are enforceable law. Pendimethalin applications are not to be made until after the alfalfa reaches the 2nd trifoliate stage, which takes place well after germination. Rhizobial inoculation takes place well before planting as a seed treatment in most cases. Even with uninoculated seed, if alfalfa has been grown in the field during the previous 3-5 years, natural inoculation occurs as soon as the seed is planted and the ‘infection’ for the symbiotic relationship is established with nodule development very soon after germination. Post-emergence inoculations are rarely attempted (generally not needed) and are known to be rarely successful.

Lines 376 & 377: ‘flanonoid concentrations’

Line 380: Replace ‘higher’ with ‘greater’

Line 381: Replace ‘increment’ with ‘increase’

Line 384: ‘Organic acid exudation . . .’

Line 389-390: ‘. . .root exudate composition could also be related to an alteration . . .’

Line 393: ‘In conclusion, pendimethalin and clethodim applications . . .’ Only base conclusions on your results and your results are limited to only two herbicides, both of which were not applied in the best manner for crop protection. It would have been appropriate for you to review the literature, including other studies published in much more widely recognized international journals than you have cited, as well as product labels from countries with more stringent regulations on pesticide use so that you would use the best product use techniques to evaluate them for crop safety. Stating that you relied on the label approved for use in Spain was not the best method to apply the product, without reviewing other labels, which are available online.

Line 400: ‘P. vulgaris’ Do not begin sentences with abbreviations, acronyms, or numerals. If revising the sentence cannot be done, spell out the term. In this case, simply spelling out the genus is fine. Otherwise, usually, adding an article (a, an, the) resolves the issue.

Line 401: Please provide the manufacturer and location for Tween 20.

Line 402: In decimal fractions, change all commas to a decimal (see Line 414, where 2.5 is used).

Line 403: Verify the spelling; should it be ‘perlerlite’ or ‘perlite’?

Line 411: ‘. . . in the laboratory . . .’

Line 415: No need to show the EC units in parentheses; that will be in the table or figure where the results are presented. Delete such units elsewhere unless reporting data at which time the units are necessary.

Line 416: Please show the manufacture and location for the Eutech CON700.

Lines 417-418 and elsewhere: There is no need to define chemical symbols. “. . . measurement of CaCO3 by . . .’ Subscript the 3. Also, convert all proportions to SI units. % should be reported as either g kg-1 or mg kg-1. Convert Table 1 data as needed.

Lines 420-421: ‘. . . using an elemental analyzer (TruSpec CN LECO®, St. Joseph, MI USA).’ Place company names within the parentheses; when showing city, state/province nation, no comma is used between the state/province and nation.

Table 1 title: ‘Soil properties: Electrical conductivity (EC); available P (Pav); CaCO3; organic C (OC); total N and total C.’ Units are in the column headings. Be sure to convert % to SI units and replace commas used in decimal fractions.

Lines 427-429: Show the manufacturer name and location in parentheses following the product. If citing the labels on Line 429, follow the same format for the citations and references as for other references. I checked the Centurion label; it really does not give much information about application timing, at least for alfalfa, which is too bad. The Ordago label was in Spanish, which I do not understand.

Line 432: ‘. . . used and each product was applied . . .’

Line 434: ‘. . . of the non-control treatments (clethodim and pendimethalin) were carried out . . .’

Line 438: Describe the irrigation/watering regime, amount and frequency during this phase of the study.

Lines 441-442: ‘Samples of shoots, roots, and nodules were weighed fresh and then dried at 70 ºC for 24 h to determine their dry weight (DW)’

Line 458: ‘. . . bean root exudates (RE), germinated seed were . . .’

Line 463: ‘preserving roots of light’ Please clarify.

Line 465: Please describe the modifications or add ‘. . . [50], with some modifications as follows: The nutrient . . .’ if that is the case.

Line 468: ‘Corp., Bellefonte, PA USA).’

Lines 470-471: Please show the manufacturer and location (city, state/provence country).

Line 474: Please provide the manufacture and location for the ACQUITY H CLASS. If the same unit was used as mentioned at lines 470-471. Show the information at that point.

Line 476: Delete comma after ‘MA’. Define ESI.

Lines 478-479: Note that the A precedes water, but B follows HiPerSolv. It seems that the B should precede HiPerSolv. Provide manufacturer and location for HiPerSolv.

Line 486: ‘The contents of . . .”

Line 487: Please provide the location for Sigma. Is that the full company name?

Lines 496-497: ‘. . . (2021) cell suspension (OD600 0.03).’ Is the 1021 a strain identifier for the Sinorhizobium? Name the manufacturer and location. What is the OD600 0.03? Clarify here and where a similar term is used at Line 511.

Lines 497, 501, & 508: ‘clethodim or pendimethalin’ Check this throughout the manuscript as I may have missed some occurrences.

Line 498: What is the basis for these concentrations

Line 500: Please provide the manufacturer and location for TY medium or describe its composition, if you mixed it.

Line 505: ‘. . . verified at 2 different times . . .’

Line 511: How long after germination did the inoculation take place? What do you mean by late exponential phase?

Line 521-520: ‘. . . with herbicide treatments (untreated control, clethodim, or pendimethalin) as the variable . . .’

Line 522: Replace ‘between’ with ‘among’ since there are three treatments in the comparison. Also, Duncan is not an LSD; it is a multiple range test. So delete LSD.

References: Check the manuscript template and Instructions to Authors for correct formatting of references. None are correctly formatted and it is not the journal staff responsibility to do that for you.

Line 551: The last digit of the year is not bold.

Line 582 and elsewhere: Italicize all scientific names.

Lines 594-595 & 647-648 & 649: Translate these references and components thereof into English and for any article not in English, state the language and whether it is accompanied by an English abstract.

Lines 607-608: This is an incomplete reference. Please include all information necessary as shown in the reference examples.

In my opinion, although the topic is of international interest and the results well-accepted by a segment of the scientific and general community that shares the views of the authors, as it stands, the information collected during this study is limited in potential use to encouraging the Ministry of Environment in Spain to be more diligent in labeling laws that include information about environmental and crop safety.  This is really too bad, but it is incumbent upon scientists to thoroughly study the available literature, include instructions about product use before conducting the study. It also is incumbent upon scientists to not include editorial information in the scientific literature that shows their bias. Both of these are great enough concerns to reject the manuscript.

Reviewer 3 Report

This manuscript entitled “Reduction in the use of herbicides favors nitrogen fixation efficiency in legumesdescribes the effect of two commonly used herbicides (pendimethalin and clethodim) on the legume-rhizobia spp. Symbiosis by detecting the plant growth and nitrogen fixation, root exudates composition, nodulation kinetics and bacterial growth and motility, and so on. This study is interesting and has certain scientific significance. There are, however, some issues that could be addressed to improve the manuscript.

1. I suggest the authors to reorganize the Introduction part (describe in several paragraphs, action mechanism of the herbicides, research progress on the effects of herbicides on crop growth and development, and so on).

2. In the Results, Authors had better add the phenotype pictures of plants for Figure 1, Figure 2 and Figure 3.

3. Figure 3 showed that pendimethalin inhibited nodule development rather than nodulation, Authors had better add the qPCR assay for representative nodulin genes  in the control and treated roots (for example, NIN, ENOD40, Nodulin35, Lb,  Apyrase GS52, and so on).